# LaRI: Layered Ray Intersections for Single-view 3D Geometric Reasoning

**Rui Li**[1]  **Biao Zhang**[1]  **Zhenyu Li**[1]  **Federico Tombari**[2,3]  **Peter Wonka**[1]

https://ruili3.github.io/lari

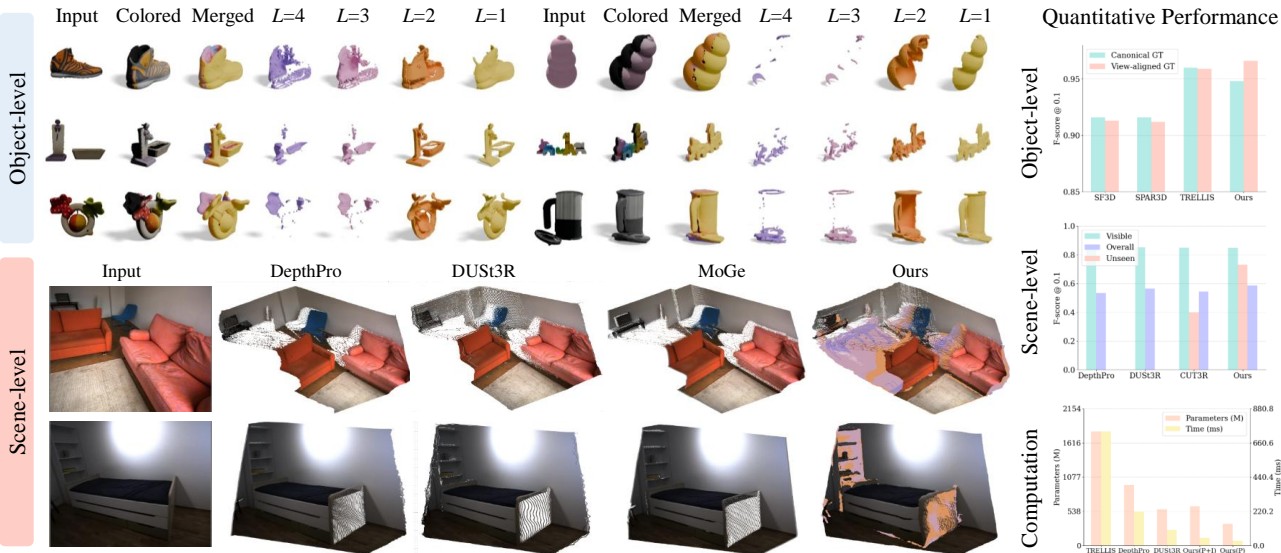

*Figure 1.* **Layered Ray Intersection (LaRI)** models multiple 3D surfaces from a single view by representing ray-surface intersections into depth-ordered, layered 3D point maps (color for different layers: 1 , 2 , 3 , 4 , . . . ). This enables a unified reconstruction recipe for object-level and scene-level tasks, leading to higher reconstruction accuracy with reduced computational overhead.

## Abstract

We present Layered Ray Intersections (LaRI), a fully supervised method for occluded geometry reasoning from a single image. Unlike conventional depth estimation, which is limited to visible surfaces, LaRI predicts multiple surfaces intersected by the camera rays using layered point maps. Compared to the existing approaches that leverage neural implicit representations or iterative refinement, LaRI achieves complete scene reconstruction in one feed-forward pass, enabling efficient and view-aligned geometric reasoning to underpin both object-level and scene-level tasks. We further propose to predict the ray stopping index, which identifies valid intersecting pixels and layers from LaRI's output. To better under-

pin and evaluate this task, we build an annotation pipeline using rendering engines, construct annotations for five public datasets, including synthetic and real-world data covering 3D objects and scenes. As a generic method, LaRI's performance is validated in object-level and scene-level reconstruction tasks.

## 1. Introduction

The natural world comprises complex 3D structures, with scenes and objects partially occluded from direct view. Nevertheless, humans excel at inferring unseen structures from available visual cues, enabling long-range navigation, collision avoidance, and interaction with environments. While replicating this ability to scene perception (Geiger et al., 2012), virtual reality (Engel et al., 2023), and robotics (Mohammadi et al., 2023) is appealing, conventional methods such as depth estimation models (Yin et al., 2023; Hu et al., 2024a; Bochkovskiy et al., 2025; Li et al., 2023), only reconstruct observable surfaces while omitting the unseen geometry.

---
[1]KAUST, Jeddah, Saudi Arabia. [2]Google, Zurich, Switzerland. [3]Technical University of Munich, Munich, Germany. Correspondence to: Peter Wonka <pwonka@gmail.com>.

*Proceedings of the 43rd International Conference on Machine Learning*, Seoul, South Korea. PMLR 306, 2026. Copyright 2026 by the author(s).

Several methods that perceive occluded environments feature individual advantages and limitations. Prior works (Wu et al., 2017; Xu et al., 2019; Choy et al., 2016; Fan et al., 2017) perform single-view 3D reconstruction on object shapes (Chang et al., 2015), and were later enhanced by diffusion models (Liu et al., 2023b; Shi et al., 2023; Hu et al., 2024b) and high-dimensional latent representations (Hong et al., 2024; Tang et al., 2024; Xu et al., 2024). Despite high-quality results, these models typically focus on (multiple) instance-level generation, without evaluating scene-level reconstruction that includes backgrounds. Another line of research addresses scene-level reconstruction using NeRF-based 3D point querying (Kulkarni et al., 2022; Li et al., 2024a), human-interactive segmentations (Li et al., 2025a), or querying with additional virtual poses (Wang et al., 2025b; Shin et al., 2019; Duisterhof et al., 2025). However, these methods are limited by computation due to multiple queries and often rely on additional inputs that are sometimes unavailable. Though novel-view synthesis approaches have shown promising results in rendering images from occluded views (Szymanowicz et al., 2024; 2025; Shih et al., 2020), they mainly focus on photorealism instead of 3D geometric quality. In light of these limitations, we consider a new model that *simultaneously adapts to object and scene tasks, being simple and efficient, while staying dedicated to 3D accuracy*.

We approach the above goal with layered ray intersections (LaRI), a simple yet effective approach to model visible and unseen 3D surfaces with multi-layer point maps, all in one feed-forward pass. We draw inspiration from layered depth images (LDIs) (Shade et al., 1998) in computer graphics and reformulate unseen geometry estimation as a multi-layer regression problem. Specifically, unlike traditional depth models that predict only the first ray-surface intersections, LaRI models all surfaces intersected by the camera ray in a depth-ordered manner. Regarding LaRI as a standard regression task, we enable unseen geometry estimation compared to traditional depth estimation, as well as view-aligned, compact 3D modeling compared to many generative models, leading to a unified approach for object- and scene-level tasks.

As LaRI's output defines a fixed maximum number of intersections for all pixels, it is necessary to identify only the valid intersection layers for each pixel. We therefore estimate the index of the layer that contains the last valid intersection, which we name the ray stopping index. We empirically verify its effectiveness over direct mask regression.

Considering the lack of data for training and evaluating this task and the model, we create annotations from 5 public datasets (Deitke et al., 2023; Fu et al., 2021; Downs et al., 2022; Yeshwanth et al., 2023; Jung et al., 2024), including both synthetic and real-world data, by combining 3D assets with existing geometry rendering engines (Community, 2018; Ravi et al., 2020).

We demonstrate the effectiveness of LaRI across domains: in object-level comparisons (Downs et al., 2022), under canonical ground truth evaluation, LaRI yields comparable results to the popular generative model (Xiang et al., 2025) using fewer parameters with $\times 10$ faster inference. Moreover, it outperforms existing methods evaluated by view-aligned ground truth. In scene-level evaluation, LaRI achieves comparable or better overall scores compared to feed-forward foundation models, with additional capability to estimate unseen surfaces. Our contributions are:

- A simple, single-view geometric model to estimate both visible and unseen surfaces in one feed-forward pass, enabling efficient and complete geometric modeling in both object and scene reconstruction tasks. Meanwhile, a ray-stopping index network is proposed for identifying valid intersections.

- A complete data annotation pipeline and evaluation benchmark to motivate further investigations on geometric reasoning about unseen surfaces.

## 2. Related Work

We review existing representations and methods related to geometric reconstruction and reasoning.

**Depth-related representation.** Depth estimation (Eigen et al., 2014) infers pixel-wise distance along the **z**-axis. With view-aligned output, it continuously benefits from advanced 2D neural networks (He et al., 2016; Dosovitskiy et al., 2020; Wang et al., 2020; Oquab et al., 2023), and leads to superior generalization abilities (Yang et al., 2024; Hu et al., 2024a; Guizilini et al., 2023; Bochkovskiy et al., 2025). It supports metric distance estimation (Hu et al., 2024a; Yin et al., 2023; Bochkovskiy et al., 2025), 3D reconstruction (Yin et al., 2021; Xu et al., 2023), as well as downstream tasks (Bhat et al., 2024; Lao et al., 2024; Xu et al., 2025; Müller et al., 2024). Recently, point-map representation (Wang et al., 2024a) has extended depth by modeling **xyz**-axes of the scene, directly supporting 3D reconstruction ranging from single-view (Wang et al., 2025c; 2024a), multi-view (Wang et al., 2024a; Leroy et al., 2024; Wang & Agapito, 2025), to dynamic scenes (Li et al., 2025b; Jin et al., 2025; Zhang et al., 2025). Our method benefits similarly from view-aligned outputs, but enables a new capability to model unseen geometry.

**Coordinate-based representation.** Prevailing approaches leverage occupancy (Mescheder et al., 2019) or signed distance function (Park et al., 2019) to represent 3D by per-point querying. Recent methods combine them with rendering techniques, such as neural radiance fields

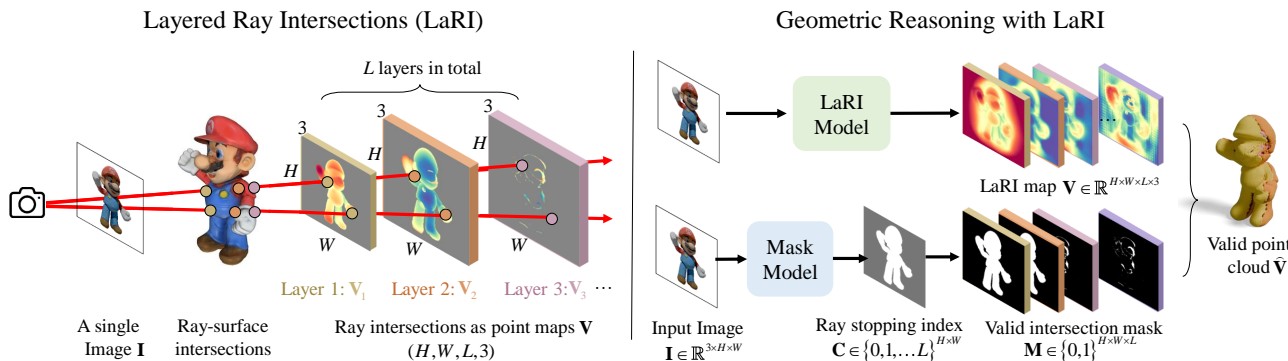

Figure 2. **Overview.** *Left*: Given an input image, the 3D geometry can be represented by the intersection points between camera rays and object surfaces. While conventional depth estimation methods only model the first intersection (i.e., layer $\boxed{1}$ ), LaRI represents all intersections (e.g., layer $\boxed{1}$ , $\boxed{2}$ , $\boxed{3}$ , . . . ) with layered 3D point maps. *Right*: Given an input image $\mathbf{I}$, the model predicts the LaRI map $\mathbf{V} \in \mathbb{R}^{H \times W \times L \times 3}$, which represents all possible intersections with a fixed layer number. It further identifies the valid ray intersections by regressing the ray stopping index $\mathbf{C} \in \mathbb{R}^{H \times W \times L}$, which is transformed into binary masks $\mathbf{M}$ to derive the final point cloud $\hat{\mathbf{V}}$.

(NeRF) (Mildenhall et al., 2020). Latest approaches (Xiang et al., 2025; Huang et al., 2025; Boss et al., 2025; Li et al., 2024b; Voleti et al., 2024) apply this representation into pre-trained models (Oquab et al., 2023; Rombach et al., 2022; Ho et al., 2022) for object-level reconstruction. Our method avoids the time-consuming per-point querying of these models and yields view-aligned results.

**Grid-based representation.** Pioneering works (Choy et al., 2016; Riegler et al., 2017) leverage 3D grid to represent object geometry. Recent advances focus on representing the scene as occupancy for autonomous driving (Miao et al., 2023; Tong et al., 2023; Cao & de Charette, 2022; Wei et al., 2023; Huang et al., 2023). Compared to these works, our method avoids the grid-level modeling at cubic cost, and is not bounded by the pre-defined 3D resolution.

**Layered representation.** In computer graphics (CG), layered depth images (LDIs) (Shade et al., 1998) represents scenes as layered depth maps for efficient image-based rendering, which is extended by layered depth cubes (Pfister et al., 2000) and depth peeling (Liu et al., 2009a;b). Our approach is inspired by LDIs but formulates single-view occluded 3D geometry reasoning as a supervised layered ray-intersection prediction problem, accompanied by ray-stopping regression and newly constructed datasets for effective learning. The concept of LDIs is also used in novel view synthesis (Shih et al., 2020; Tulsiani et al., 2018; Szymanowicz et al., 2025; 2024) or optical flow estimation (Wen et al., 2024) for transparent regions. Our method focuses specifically on 3D geometry, with dedicated geometry supervision to ensure 3D fidelity.

**Occluded geometry estimation.** Existing methods for complete object reconstruction rely on large generative 3D models (Xiang et al., 2025) or video models (Hu et al., 2024b), yet their efficiency is limited and their efficacy in complex scenes has not been verified. Scene-level methods usually adopt a multi-iteration scheme, with potentially additional

inputs. Semantic scene complete methods (Huang et al., 2024; Wu et al., 2020) infer missing regions from RGB-D scans than images, with a progressive refinement scheme. KYN (Li et al., 2024a) and DRDF (Kulkarni et al., 2022) estimate unseen geometry by dense 3D coordinate queries, significantly slowing down inference speed. AmodalDepth estimation (Li et al., 2025a) requires human interactive amodal segmentations, while CUT3R (Wang et al., 2025b), MLD (Shin et al., 2019), and Rayst3R (Duisterhof et al., 2025) necessitate additional virtual pose as queries, which is non-trivial to acquire. In a word, existing methods either require non-trivial heuristics or perform multiple iterations. As a contrast, our method operates on a single image in only one feed-forward pass, without using other heuristics.

## 3. Method

### 3.1. Layered Ray Intersections as Point Maps

To enable geometric reasoning for unseen surfaces, our method represents the complete geometric structure by ray-surface interactions. This concept is related to layered depth images (LDIs) (Shade et al., 1998) in computer graphics, where existing 3D geometry is represented using multiple depth layers for efficient image rendering. However, unlike the natural world, where light rays stop once they first intersect an opaque surface, we hypothesize rays as intersecting the various surfaces they meet along their path. This approach diverges from both real-world light behavior, where rays may interact with multiple surfaces via reflection or refraction, and standard rendering pipelines, where occluded surfaces are ignored. By explicitly estimating all intersection points, the occlusion-aware geometric model enables direct recovery of unseen surfaces along camera rays.

**Layered ray intersection map.** We model the LaRI representation using point maps, i.e., the layered 3D coordinate maps representing the intersection positions between

each ray and the surfaces. Importantly, all intersections are recorded, including those with occluded surfaces. As shown in Fig. 2 (*Left*), given an image $\mathbf{I} \in \mathbb{R}^{H \times W \times 3}$ of height $H$ and width $W$, each of these pixels is associated with a camera ray. A map $\mathbf{V}_l \in \mathbb{R}^{H \times W \times 3}$ stores the $l$-th intersection point of the rays intersecting the object/scene. We stack all the $L$ maps into one single tensor $\mathbf{V} \in \mathbb{R}^{H \times W \times L \times 3}$, termed the LaRI map, where $L$ denotes the maximum number of intersection layers. We aim to estimate the LaRI map $\mathbf{V}$.

**Ray intersection mask.** While the LaRI map uses a fixed number of layers, the actual ray-surface intersection number for each pixel varies. For example, object-level images usually contain background areas with no ray intersection at all, and the indoor images can contain regions with a single intersection (e.g., a wall) or multiple intersections (e.g., a chair in front of a wall). It is difficult to represent these invalid intersections in the LaRI map, as these intersections exhibit infinite distances that are hard to regress. To this end, we introduce an additional mask $\mathbf{M} \in \{0, 1\}^{H \times W \times L}$ to identify the valid ray intersections per pixel across layers. We can then derive the resulting valid 3D point cloud by querying LaRI map $\mathbf{V}$ with the mask $\mathbf{M}$,

$$\hat{\mathbf{V}} = \{\mathbf{V}(h, w, l) \,|\, \mathbf{M}(h, w, l) = 1\}, \qquad (1)$$

where $h, w, l$ denote the index of $H, W, L$.

**Relation to previous representations**. LaRI can act as a general geometric model that augments existing representations with the following advantages: (1) *Completeness*: It models both visible and unseen geometry from a single image, extending the reasoning capacity of depth-based representations (Hu et al., 2024a; Bochkovskiy et al., 2025; Wang et al., 2024a). (2) *Support for single feed-forward pass*: It inferences of all ray intersections in a single feed-forward pass, avoiding the time-consuming NeRF-based/grid-based dense sampling (Li et al., 2024a; Wimbauer et al., 2023; Yu et al., 2021; Xiang et al., 2025), or mask/pose-based queries iterations (Li et al., 2025a; Yu et al., 2025; Duisterhof et al., 2025; Wang et al., 2025b). (3) *Compactness*: It only involves intersection points between ray and surfaces, circumventing modeling the large freespace as in NeRF (Li et al., 2024a; Yu et al., 2021), occupancy models (Wang et al., 2023; Tong et al., 2023), and many generative approaches (Xian et al., 2020; Huang et al., 2025; Wang et al., 2024b). (4) *View-aligned predictions*: It predicts 3D point clouds in the camera coordinate system, like depth scanners, eliminating further post-processing steps to align 3D with the image and simplifying downstream applications.

**Overview.** As shown in Fig. 2 (*Right*), the input image is sent to the LaRI map prediction network for intersection point prediction (Sec. 3.2). To reconstruct only areas with valid ray intersections, we estimate the valid intersection mask by predicting the ray stopping index (Sec. 3.3).

Considering the lack of proper datasets to train and evaluate this task, we construct a complete annotation pipeline to construct data from 5 datasets, including synthetic 3D assets (Deitke et al., 2023; Fu et al., 2021) and real-world scans (Yeshwanth et al., 2023), using graphics engines (Ravi et al., 2020) 3.4.

### 3.2. Layered Ray Intersections Regression

As the LaRI map encodes ray intersections in a camera-aligned manner, we formulate its prediction as a multi-layer point map regression task. This allows us to leverage existing powerful 2D networks and their pre-training weights (Dosovitskiy et al., 2020; Oquab et al., 2023; Siméoni et al., 2025).

**Networks.** We adopt a generic encoder-decoder architecture popular for 2D regression tasks. Specifically, we choose the ViT-Large (Wang et al., 2004) as the backbone, and an adapted CNN-based regression network (Wang et al., 2025c; Ranftl et al., 2021) to regress layers of point maps. For each processed feature $\mathbf{F}$ through the encoder and decoder, we add dedicated heads to regress point maps for each layer

$$\mathbf{V}_l = \mathtt{head}_l(\mathbf{F}), \qquad (2)$$

$$\mathbf{V} = \mathtt{concat}(\{\mathbf{V}_l\}_{l=1}^{L}), \qquad (3)$$

where $\mathbf{V}_l$ is the point map for each layer. This simple design enables the use of priors from existing approaches and has demonstrated competitive results in object and scene reconstructions.

**Loss function.** We design the predicted LaRI map to encode relative geometry, i.e., to be transformed from the ground truth by one global scale factor and one $\mathbf{z}$-axis shift factor. Therefore, we leverage the scale-shift alignment Euclidean loss as supervision: given the network prediction $\mathbf{V}_{\mathrm{pred}}$ and ground truth $\mathbf{V}_{\mathrm{gt}}$, we perform scale-shift alignment using the least-square (Ranftl et al., 2020; Bhat et al., 2023) method:

$$\mathbf{s}^*, t^* = \arg\min_{\mathbf{s}, t} \sum_{|v|} (\mathbf{s} \cdot \hat{\mathbf{v}}_{\mathrm{pred}} + t - \hat{\mathbf{v}}_{\mathrm{gt}})^2, \qquad (4)$$

where $\mathbf{s}^*$ is the global scaling factor for all $\mathbf{x}, \mathbf{y}, \mathbf{z}$ axes of the prediction, and $t$ is the shift factor only for $\mathbf{z}$ axis. Additionally, $\hat{\mathbf{v}}_{\mathrm{pred}} \in \hat{\mathbf{V}}_{\mathrm{pred}}$ and $\hat{\mathbf{v}}_{\mathrm{gt}} \in \hat{\mathbf{V}}_{\mathrm{gt}}$ are valid prediction- and ground-truth point coordinates selected by ground truth ray intersection mask $\mathbf{M}_{\mathrm{gt}}$ according to Eq. 1. We further use a Euclidean loss to supervise the network prediction

$$\begin{aligned} \mathcal{L}_{pm} &= \|\hat{\mathbf{v}}'_{\mathrm{pred}} - \hat{\mathbf{v}}_{\mathrm{gt}}\|, \\ \hat{\mathbf{v}}'_{\mathrm{pred}} &= \mathbf{s}^* \cdot \hat{\mathbf{v}}_{\mathrm{pred}} + t^*, \end{aligned} \qquad (5)$$

where $\hat{\mathbf{v}}'_{\mathrm{pred}}$ is the scale-shifted prediction from the least-square method.

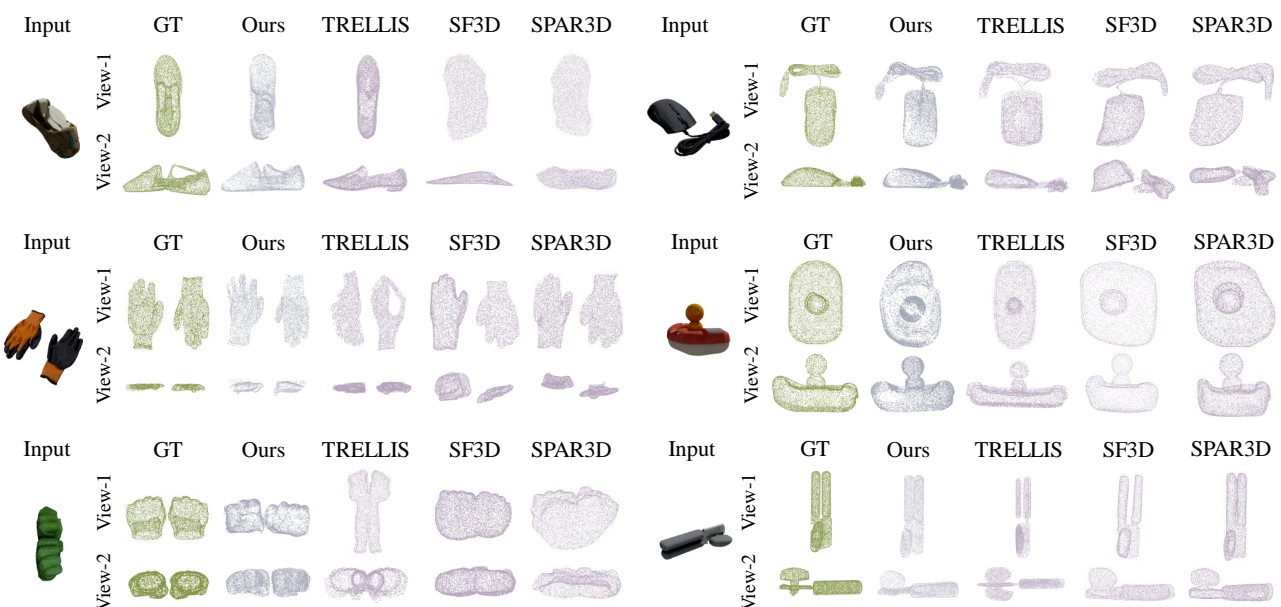

*Figure 3.* **Qualitative comparisons on GSO**. Our method predicts more faithful results to the input image compared to the large generative model.

### 3.3. Ray Intersection Mask Regression

As different rays have varying numbers of intersections to their corresponding surfaces (Sec. 3.1), a ray intersection mask $\mathbf{M} \in \{0,1\}^{H \times W \times L}$ is needed to identify the valid intersection points from the fixed-sized LaRI map $\mathbf{V} \in \mathbb{R}^{H \times W \times L \times 3}$. Training a network for predicting this mask is challenging: unlike prevailing models (Cheng et al., 2022a;b; Kirillov et al., 2023), segmenting unordered instances and classes, valid ray intersection segmentation imposes a strict order, i.e., valid indices must begin at the first layer and continue consecutively up to the last intersection. Although prior works (Fernandes & Cardoso, 2018; Diaz & Marathe, 2019; Ravi et al., 2025) explore ordinal or temporal relations in semantic/instance-level segmentation, segmenting valid ray intersections remains underexplored.

**Ray stopping index regression.** We propose a simple formulation to predict the ray intersection mask. Instead of predicting multiple layers of uncorrelated binary masks, we predict the *ray stopping index*, i.e., the last surface index that a ray travels through. This naturally enforces depth ordering by marking all layers before the stopping index as valid. Given an input image $\mathbf{I} \in \mathbb{R}^{H \times W \times 3}$, the model outputs the ray stopping logits $\mathbf{S} \in \mathbb{R}^{H \times W \times (L+1)}$, which can be transformed into ray stopping index $\mathbf{C} \in \{0, 1, \ldots, L\}^{H \times W}$ by

$$\mathbf{C}(h, w) = \arg\max_{l} \operatorname{softmax}(\mathbf{S}(h, w, l)). \qquad (6)$$

Note that the number of indices is $L + 1$, with index "0" denoting "no intersection" and the remaining indices indicating the stopping indices for each layer. During inference, the ray intersection mask $\mathbf{M} \in \{0,1\}^{H \times W \times L}$ can be derived by

$$\mathbf{M}(h, w, l) = \begin{cases} 1 & \text{if } l + 1 \leq \mathbf{C}(h, w), \\ 0 & \text{otherwise.} \end{cases} \qquad (7)$$

We adopt a separate ViT-Large backbone (Wang et al., 2004) and a dense segmentation decoder (Ranftl et al., 2021) for this task, with a cross-entropy loss to supervise the output logits

$$\mathcal{L}_{rs} = \operatorname{CrossEntropy}(\mathbf{S}_{\text{pred}}, \mathbf{S}_{\text{gt}}), \qquad (8)$$

where $\mathbf{S}_{\text{pred}}$ and $\mathbf{S}_{\text{gt}}$ are prediction and ground truth logits respectively.

### 3.4. Data Construction

One challenge to verify the LaRI model is the lack of annotated data. To address this issue, we have collected 5 datasets to enable comprehensive training and evaluation. The details can be found in Table 1. For synthetic data, we leverage Blender (Community, 2018) to render images and Pytorch3D (Ravi et al., 2020) to render multi-layer point maps with simulated trajectories. For real-world data, we use Pytorch3D (Ravi et al., 2020) to render multi-layer point maps from the given camera poses, to align with the input image perspective. Details about the rendering process can be seen in the Appendix A.1.

## 4. Experiments

### 4.1. Datasets

**Training data.** We train the object and scene model separately. For the object model, we use the Objaverse (Deitke et al., 2023) dataset, which contains around 192K images. For the scene model, we use the 3D-FRONT (Fu et al.,

*Table 1.* **Dataset statistics**.

| Datasets | Type | #Scenes | #Frames | Visible v.s. Unseen (%) | Usages |
|---|---|---|---|---|---|
| Objaverse (Deitke et al., 2023) | Synthetic | 16K | 192K | 48/52 | Training |
| 3D-FRONT (Fu et al., 2021) | Synthetic | 18K | 108K | 42/58 | Training & Evaluation |
| ScanNet++ (Yeshwanth et al., 2023) | Real | 1K | 50K | 74/26 | Training |
| SCRREAM (Jung et al., 2024) | Real | 11 | 460 | 51/49 | Evaluation |
| GSO (Downs et al., 2022) | Synthetic | 1K | 37K | 41/59 | Evaluation |

2021), including around 108K images. As 3D-Front is a synthetic dataset that may result in real-world generalization problems, we also use the ScanNet++ (Yeshwanth et al., 2023), a real-world 3D-scanned dataset containing around 48K images.

**Evaluation data.** For object-level evaluation, we use the full set of Google Scanned Objects (GSO) (Downs et al., 2022) containing 1,030 objects, with images rendered by (Liu et al., 2023b)'s script. We set elevation angles to $[0°, 30°, 60°]$ and render 12 images with even azimuth angles for each elevation, resulting in 37080 images. For scene-level evaluation, we select an individual subset of 3D-Front (Fu et al., 2021) with no overlap with the training data. Meanwhile, we also choose the SCRREAM dataset (Jung et al., 2024), a real-world indoor dataset with complete scanned meshes for all scene components (chair, table, sofa, *etc*). We sample SCRREAM video frames at an interval of 5. For both datasets, we select frames with at least 30% of the pixels containing unseen structures, leading to 460 SCRREAM images and 2300 3D-FRONT images. As Scan-Net++ contains incomplete unseen regions, we mainly adopt its partial non-training images for visualizations.

### 4.2. Evaluation and Metrics

We adopt 3D evaluation metrics, i.e., Chamfer Distance (CD) and the F-score (FS) of different distance thresholds (e.g., 0.1, 0.05, 0.02), which are used by both 3D reconstruction (Liu et al., 2023a) and depth estimation methods (Örnek et al., 2022; Spencer et al., 2024). Please refer to Appendix A.2 for details.

**Object-level evaluation.** We evaluate under two ground truth (GT) settings: (1) *Canonical GT*. The GT is expressed in a canonical coordinate system, an adopted practice for evaluating object-level generative methods (Hu et al., 2024a; Bochkovskiy et al., 2025; Wang et al., 2024a; 2025c), since the output coordinates are usually unknown. All methods must go through the brute-force search with ICP registration (Chetverikov et al., 2002). (2) *View-aligned GT*. The GT is uniformly sampled and is transformed to the camera view, leading to a pixel-aligned evaluation that is widely adopted by depth estimation methods (Hu et al., 2024a; Bochkovskiy et al., 2025; Wang et al., 2024a; 2025c). For object-level evaluation, we use the canonical GT points that are uniformly sampled from the mesh surfaces. We trans-

form the canonical GT points using camera poses during view-aligned evaluation. We sample 10,000 points for all methods.

**Scene-level evaluation.** We report results for visible, unseen, and combined visible and unseen surfaces (overall). We used the GT points from the layered representation, which allows us to evaluate the visible and unseen regions individually. Note that we use 10 layers of GT point maps to ensure a broader coverage than the predicted layers (which is 5). The layer-coverage statistics are in Figure 6. For depth estimation models, we convert the depth maps into a 3D point cloud with predicted focal lengths. Predictions and GT are aligned using scale-shift alignment as reported in Eq. 4. We sample 100,000 points for all methods and use the GT mask for evaluation.

### 4.3. Implementation Details

We use ViT-Large (Dosovitskiy et al., 2020) as the backbone with pre-trained weights from (Wang et al., 2025c). We train the model using PyTorch (Paszke et al., 2017), with AdamW (Loshchilov & Hutter, 2017) as the optimizer. We train our object-level model and scene-level model separately: for object-level training, we use a learning rate $10^{-4}$ and 10 epochs for cosine warm-up; For scene-level training, we use a learning rate $10^{-4}$ and 5 warm-up epochs. We train models with a total batch size of 96 using 4 NVIDIA A100 80G GPUs. The input resolution is fixed to $512 \times 512$. We randomly crop the image for indoor data, and align it to the training resolution by resizing the long side to 512, and complementing the short side with the default, gray color.

### 4.4. Object-level Comparison

We compare LaRI with existing methods for object-level single-image generation or reconstruction, including the image-supervised method (Boss et al., 2025), point cloud supervised method (Huang et al., 2025), and the depth/normal supervised method (Xiang et al., 2025).

As shown in Table 2, with canonical GT, all results are registered using brute-force search and ICP. Our method outperforms existing methods (Huang et al., 2025; Boss et al., 2025) trained on the same Objaverse by notable margins, with FS@0.02 improving by 18% over the latest SPAR3D (Huang et al., 2025). Meanwhile, our method yields slightly inferior performance compared to TREL-

*Table 2.* **Object-level comparison on the GSO dataset**. We show results with ground truth in canonical and camera coordinates.

| Method | Canonical Ground Truth | | | | View-aligned Ground Truth | | | |
| --- | --- | --- | --- | --- | --- | --- | --- | --- |
| | CD ↓ | FS@0.1 ↑ | FS@0.05 ↑ | FS@0.02 ↑ | CD ↓ | FS@0.1 ↑ | FS@0.05 ↑ | FS@0.02 ↑ |
| SF3D (Boss et al., 2025) | 0.036 | 0.916 | 0.754 | 0.513 | 0.037 | 0.913 | 0.738 | 0.487 |
| SPAR3D (Huang et al., 2025) | 0.037 | 0.916 | 0.759 | 0.506 | 0.038 | 0.912 | 0.745 | 0.486 |
| TRELLIS (Xiang et al., 2025) | 0.027 | **0.960** | **0.856** | **0.611** | 0.027 | 0.959 | 0.853 | 0.608 |
| DUSt3R (Wang et al., 2024a) | - | - | - | - | 0.220 | 0.324 | 0.240 | 0.153 |
| VGGT (Wang et al., 2025a) | - | - | - | - | 0.332 | 0.335 | 0.274 | 0.200 |
| MoGe (Wang et al., 2025c) | - | - | - | - | 0.160 | 0.373 | 0.275 | 0.192 |
| Ours | 0.029 | 0.948 | 0.840 | 0.601 | **0.025** | **0.966** | **0.894** | **0.643** |

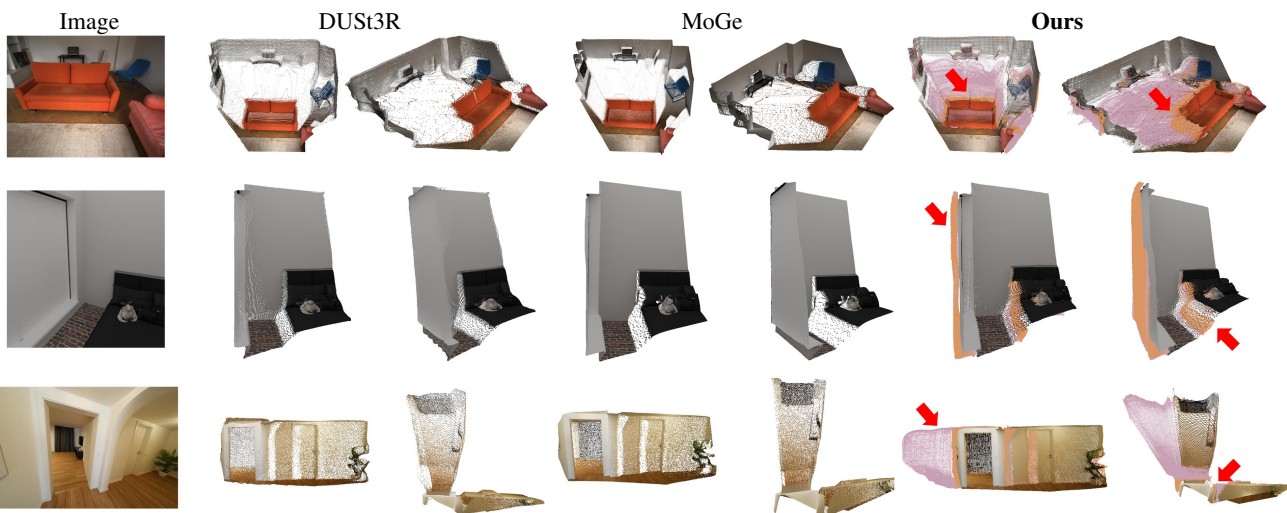

| Image | DUSt3R | MoGe | **Ours** |

*Figure 4.* **Qualitative comparisons in scene-level reconstruction**. For LaRI, we highlight different unseen layers with colors (e.g., layer 2 3 ). Compared to methods that focus only on visible surface reconstruction, our method extends the modeling coverage by reasoning about unseen regions, such as the occluded floor, sofa (row 1), walls, bed (row 2), or unseen room spaces (row 3).

LIS (Xiang et al., 2025), using fewer parameters, as shown in Table 6. In the view-aligned GT evaluation, LaRI's output inherently aligns to the camera view, which is useful for downstream tasks (e.g., tasks requiring RGB-D), without additional alignment between 3D shapes and input views. As a result, it outperforms competing methods, with 6% improvement in FS@0.02 compared to TRELLIS and 32% improvement over SPAR3D. Note that scene-level reconstruction methods, such as VGGT (Wang et al., 2025a), exhibit limited performance on object-level tasks, primarily because they focus on recovering only observable surfaces and therefore produce incomplete geometry.

Qualitative results are shown in Fig. 3. Our method shows higher visual quality than existing methods trained on the same Objaverse dataset (Huang et al., 2025; Boss et al., 2025). Note that the large model TRELLIS (Xiang et al., 2025) exhibits excellent visual quality with complete and plausible shapes. However, its results are not always faithful to the input image, e.g., as shown in the first instance, TRELLIS produces a high-quality low-heel shoe which is inconsistent with the type shown in the input image. Instead, our method is deterministic, and its estimated geometry is faithful to the input image, yielding better accuracy.

## 4.5. Scene-level Comparison

We compare LaRI with single-feed-forward methods that support point cloud output in Tab. 3. Main competing methods include metric depth estimation methods (Hu et al., 2024a; Bochkovskiy et al., 2025) and point map-based methods (Wang et al., 2024a; 2025c). As current unseen geometry estimation methods (Wang et al., 2025b; Li et al., 2025a) require human interaction or prior information (e.g., poses, masks) that limit real-world applicability, here we only evaluate one method, CUT3R (Wang et al., 2025b), for reference. We query the model with five more virtual poses (iter-5) sampled by adding noise to the original input view.

We compare different methods in "Visible", "Unseen", and "Overall" regions, respectively. Our method yields moderate performance in visible surfaces compared to the geometric foundation models (Hu et al., 2024a; Wang et al., 2025c; Bochkovskiy et al., 2025). However, our method enables unseen surface reasoning, leading to a higher level of completeness than existing methods. As a result, our method achieves better final overall scores over DepthPro (Bochkovskiy et al., 2025) and DUSt3R (Wang et al., 2024a), leading to competitive or better performance than the strong single-view 3D

*Table 3.* **Scene-level comparison across datasets**. We report Chamfer Distance (CD, ↓) and F-score@0.05 (↑) for *Visible*, *Unseen*, and *Overall* regions. LaRI achieves the best performance in unseen regions while maintaining competitive or better performances in visible and overall regions.

| | 3D-FRONT | | | | | | SCRREAM | | | | | |
| | Visible | | Unseen | | Overall | | Visible | | Unseen | | Overall | |
| Method | CD↓ | FS@0.05↑ | CD↓ | FS@0.05↑ | CD↓ | FS@0.05↑ | CD↓ | FS@0.05↑ | CD↓ | FS@0.05↑ | CD↓ | FS@0.05↑ |
|---|---|---|---|---|---|---|---|---|---|---|---|---|
| Metric3D-v2 (Hu et al., 2024a) | 0.252 | 0.118 | – | – | 0.279 | 0.123 | 0.063 | 0.534 | – | – | 0.086 | 0.473 |
| DepthPro (Bochkovskiy et al., 2025) | 0.050 | 0.696 | – | – | 0.103 | 0.562 | 0.055 | 0.603 | – | – | 0.079 | 0.535 |
| DUSt3R (Wang et al., 2024a) | 0.061 | 0.651 | – | – | 0.116 | 0.525 | 0.059 | 0.653 | – | – | 0.086 | 0.565 |
| MoGe (Wang et al., 2025c) | 0.040 | 0.788 | – | – | 0.096 | 0.621 | 0.035 | 0.786 | – | – | 0.063 | 0.668 |
| CUT3R (Wang et al., 2025b) (iter-5) | 0.093 | 0.397 | 0.271 | 0.164 | 0.146 | 0.314 | 0.071 | 0.658 | 0.192 | 0.238 | 0.091 | 0.543 |
| **Ours** | 0.050 | **0.839** | **0.076** | **0.739** | **0.061** | **0.799** | 0.057 | 0.589 | **0.077** | **0.494** | **0.059** | 0.590 |

*Table 4.* **Ablation studies on the number of layers** $L$. Object-level data is more sensitive to the layer number.

| $L$ | GSO | | | SCRREAM | | |
| | CD↓ | FS@0.1↑ | FS@0.05↑ | CD↓ | FS@0.1↑ | FS@0.05↑ |
|---|---|---|---|---|---|---|
| 3 | 0.072 | 0.752 | 0.527 | 0.061 | 0.822 | **0.590** |
| 5 | **0.025** | 0.966 | **0.894** | **0.059** | 0.825 | **0.590** |
| 8 | 0.027 | **0.967** | 0.882 | 0.061 | 0.813 | 0.575 |

*Table 5.* **Ablation studies on pre-trained weights**. DINO-v2 weights lead to comparable performance in object reconstruction, while falling short in the scene-level data.

| Pre-training | GSO | | | SCRREAM | | |
| | CD↓ | FS@0.1↑ | FS@0.05↑ | CD↓ | FS@0.1↑ | FS@0.05↑ |
|---|---|---|---|---|---|---|
| No weights | 0.070 | 0.756 | 0.506 | 0.137 | 0.534 | 0.319 |
| DINOv2 (Oquab et al., 2023) | 0.025 | 0.967 | 0.893 | 0.064 | 0.800 | 0.560 |
| DINOv3 (Oquab et al., 2023) | **0.024** | **0.972** | **0.910** | 0.064 | 0.805 | 0.565 |
| MoGe (Wang et al., 2025c) | 0.025 | 0.966 | 0.894 | **0.059** | 0.825 | 0.590 |

reconstruction method MoGe (Wang et al., 2025c). Qualitative results are in Fig. 4, the colored points indicate geometry from different layers. Our method reasons about unseen geometry for the complete scene, including background regions (floor, walls) and foreground objects (sofa, bed), leading to extended perception coverage. More visualizations can be found in Fig. 8.

### 4.6. Ablation studies

We perform detailed investigations on how the major components, key hyperparameters, and contributions of LaRI affect the final results.

**Number of layers.** As the number of layers $L$ of the LaRI map is crucial for unseen geometric reasoning, we set $L$ to $\{3, 5, 8\}$ to investigate its influence on the final performance. As shown in Table 4, object data is more sensitive to the number of layers compared to scene-level data, possibly because the object data contains a higher ratio of unseen surfaces due to self-occlusion. For both data types, the performances are close, and we choose $L = 5$ as the default setting.

**Pre-trained weights.** We run our method (1) with no pre-trained weights, (2) with DINO-v2 (Oquab et al., 2023) and DINO-v3 (Siméoni et al., 2025) weights, and (3) with weights from a 3D point map estimation method (Wang et al., 2025c). As shown in Table 5, pre-training is crucial to our method. The model with DINO-v2/v3 weights yields similar or better results than the geometric model's weights. Meanwhile, weights from the geometric model perform better in scene-level data. This indicates that LaRI can benefit widely from different pre-trained priors.

**Efficiency.** We evaluate both network parameters and infer-

*Table 6.* **Comparisons in efficiency**. LaRI is significantly smaller and faster compared to generative models, while being comparably efficient to other feed-forward methods.

| Methods | Params (M) | Time (ms) |
|---|---|---|
| SF3D (Boss et al., 2025) | 1006.0 | 123.1 |
| SPAR3D (Huang et al., 2025) | 2026.3 | 904.8 |
| TRELLIS (Xiang et al., 2025) | 1795.7 | 733.7 |
| **Ours** (point map + index models) | 620.2 | 51.5 |
| **Ours** (point map model) | 314.2 | 31.5 |
| DepthPro (Bochkovskiy et al., 2025) | 951.9 | 220.3 |
| DUSt3R (Wang et al., 2024a) | 571.1 | 100.1 |
| MoGe (Wang et al., 2024a) | 341.2 | 41.08 |
| **Ours** (point map + index models) | 620.2 | 51.5 |
| **Ours** (point map model) | 314.1 | 31.5 |

ence speed. As shown in Table 6, we report the computation efficiency for (1) the point map prediction model alone and (2) the combined setup including both the point map model and the stopping index model. Under both settings, our model contains fewer parameters and achieves faster inference for object-level reconstruction compared to existing large single-view or generative models (Boss et al., 2025; Xiang et al., 2025; Huang et al., 2025). For scene-level comparison, our point map estimation model attains efficiency comparable to or better than all existing depth- or point–map–based methods, while additionally providing reasoning over unseen layers. When including the stopping index predictor, the overall model remains similar in parameter count and runtime to DUSt3R and DepthPro, though it is less efficient than MoGe.

**Ray intersection mask.** We evaluate our ray intersection index prediction against direct binary mask regression using standard segmentation metrics. As shown in Table 7, our

*Table 7.* **Comparison of mask prediction strategies**. Ray stopping index regression demonstrates notable improvements over the binary segmentation in both object and scene reconstruction tasks.

| Mask prediction type | GSO | | SCRREAM | |
|---|---|---|---|---|
| | mIoU | DICE | mIoU | DICE |
| Binary segmentation | 0.091 | 0.154 | 0.231 | 0.275 |
| Ray stopping index | **0.560** | **0.594** | **0.546** | **0.635** |

method (Eq. 6, Eq. 7) outperforms the binary strategy by explicitly modeling the ray intersection process.

## 5. Limitations and Conclusions

**Conclusions.** We present a new approach for single-view geometric reasoning in one feed-forward pass. By representing the unseen geometry with layered intersections of rays and surfaces, our method allows for view-aligned, complete geometric reasoning efficiently. This unifies object- and scene-level reconstruction and demonstrates notable improvements over existing methods.

**Limitations.** As shown in Fig. 5, our method produces lower point density in surfaces parallel to the camera ray, or areas between layers. This is due to the inher-

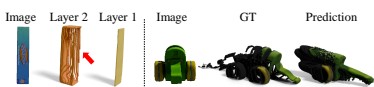

*Figure 5.* **Limitations**. (1) Lower point density on areas between layers or surfaces that are parallel to camera rays; (2) Inaccurate modeling against limited observations.

ent limitation of layered intersections and could be alleviated by post-processing. As a deterministic approach, our method can produce inaccurate results when observed information shows limited context for occluded parts. Our dataset is still limited in diversity and scale, and part of our training data is generated through synthetic rendering pipelines, which may introduce a moderate domain gap to real-world observations. We plan to extend the data for outdoor scenes in future work.

## Impact Statement

This work advances single-image 3D perception by enabling efficient reasoning about occluded geometry through layered ray intersections and a ray-stopping index. Improving the completeness and view-aligned accuracy of 3D reconstruction can benefit downstream applications such as robotics, AR/VR, and general scene understanding, where partial observability is common.

We encourage deployment with appropriate safeguards: uncertainty-aware integration, e.g., conservative planning when occluded geometry is ambiguous, thorough evaluation on target domains and diverse environments, and careful documentation of dataset composition and failure modes.

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

# A. Appendix

## A.1. Data Construction and Processing

Considering the lack of annotated data for training and evaluating LaRI, we build a data curation pipeline by carefully organizing synthetic 3D assets, well-scanned real-world data (Yeshwanth et al., 2023), and modern rendering engines (Ravi et al., 2020; Community, 2018), with data pre-processing and filtering steps to underpin faithful geometric reasoning.

**Objaverse annotation.** We use Blender (Community, 2018) to render the images of the model following the steps of (Liu et al., 2023b). Then, we use Pytorch3D (Ravi et al., 2020) to render points for each intersection layer. For Objaverse data, we found that many of the objects contain random internal structures. These 3D artifacts yield noisy and unpredictable LaRI maps that hinder model reasoning. To address this issue, we filter out samples with large areas of intersection after the second layer. We further filter out extremely small objects, yielding 16K valid objects. We render 12 views for each model, yielding 190K annotated images.

**3D-FRONT annotation.** Many indoor synthetic datasets such as 3D-FRONT (Fu et al., 2021) contain house-level meshes with multiple rooms. Rendering LaRI maps from one room using a house-level mesh will lead to excessive ray intersections with other irrelevant rooms. We further process the dataset to split houses into individual rooms and select rooms with at least two furniture items for geometric diversity. This leads to 18K room-level scenes. We render six views with random textures on the walls and floors (Denninger et al., 2023), resulting in 100K annotated images. During the rendering process, we control the camera perspective to ensure at least 1 furniture is within the field of view.

**ScanNet++.** We additionally use the real-world ScanNet++ dataset to complement the synthetic 3D-FRONT dataset for improved real-world generalization. ScanNet++ (Yeshwanth et al., 2023) contains large-scale scanned meshes for indoor room-level data. We use real-world images captured by video, with LaRI maps rendered from the mesh and the given poses. To avoid high overlap between video frames, we subsample the sequence with fixed intervals, leading to 50K real-world image pairs.

**GSO.** We use Google Scanned Objects (Downs et al., 2022) (including 1030 objects) to evaluate the object model. We adopt the same rendering protocol as used in the Objaverse dataset. For each object, we render 36 images from the top sphere, leading to 37K images in total.

**SCRREAM.** The SCRREAM dataset (Jung et al., 2024) contains 11 real-world scenes with complete object- and scene-level scannings. We render the multi-layer point maps using Pytorch3D, directly using the ground truth poses given by the dataset. To reduce the redundancy from similar frames, we sample the frames with an interval of 5.

**Coordinate convention transformations.** Different datasets and rendering engines adopt varying coordinate and camera conventions, which require non-negligible engineering effort to ensure correct annotations and consistent rendering.

For example, Objaverse (Deitke et al., 2023) data is stored in `.glTF` format and must be explicitly converted to the Blender coordinate system before applying world-to-camera transformations in PyTorch3D. Camera conventions also differ across systems: Blender uses (Y-up, Z-backward, X-right), PyTorch3D uses (Y-up, Z-forward, X-left), and real-world computer vision datasets typically adopt (Y-down, Z-forward, X-right). These differences necessitate additional transformations when transferring camera poses between systems. For example, (1) when rendering GT using Pytorch3D from poses from the Blender image renderer (as used in GSO, Objaverse, 3D-FRONT), one needs to perform a transformation between the Blender camera and the Pytorch3D camera. (2) When rendering GT using Pytorch3D from poses of computer vision datasets (SCRREAM, ScanNet++), one must perform a transformation between the computer vision coordinate and the Pytorch3D coordinate.

Another subtlety is that PyTorch3D applies transformations via right-multiplication. Therefore, when performing world-to-camera conversion, one must transpose the rotation matrix relative to conventions used elsewhere. Failing to account for this leads to incorrect geometry alignment.

In addition to these camera system discrepancies, inconsistencies exist across different synthetic 3D assets. All in all, dataset construction for this task remains time-consuming. To ease reproducibility, we will release our data generation pipeline and seek more efficient strategies for scaling up dataset preparation in future work.

**Dataset Statistics.** The layered point map annotations contain both visible and unseen regions, aiming to cover the whole scene. We show the coverage ratio of the scene with respect to the number of layers for each dataset. As shown in Figure 6, most of the surfaces are in the first five layers of the point maps. Incorporating 9 layers will cover 99% of the scene surfaces.

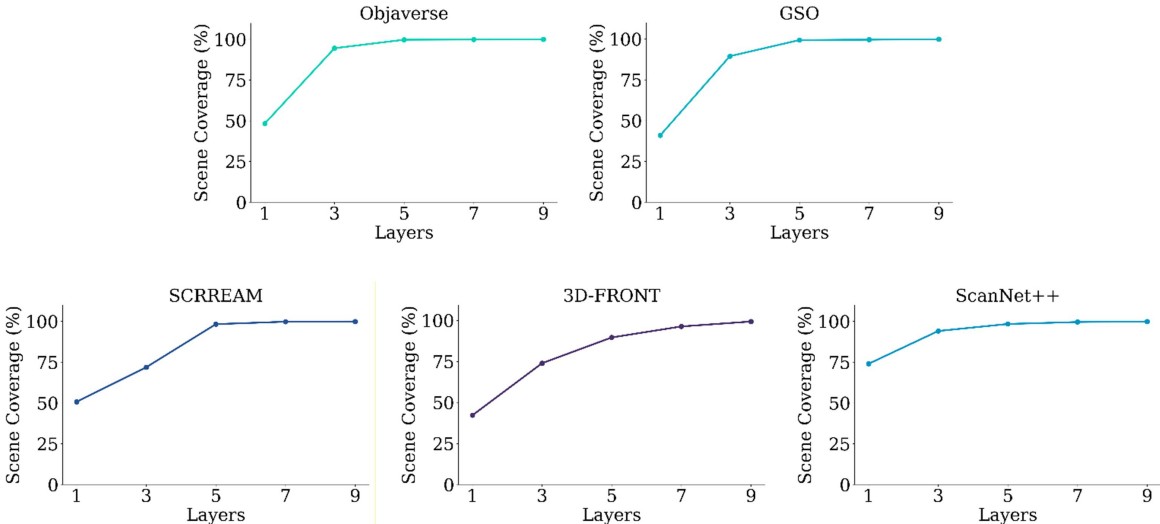

*Figure 6.* Scene coverage ratios of different annotated datasets across layers.

### A.2. Evaluation Details

**Evaluation metrics.** We adopt Chamfer Distance (CD) and F-score to evaluate the reconstruction quality of the competing methods. The chamfer distance computes the bidirectional minimal distances between the predicted 3D point cloud and the GT 3D point cloud. Given the predicted LaRI map $\mathbf{V}_{\text{pred}} \in \mathbb{R}^{H \times W \times L \times 3}$ and ground truth $\mathbf{V}_{\text{gt}} \in \mathbb{R}^{H \times W \times L \times 3}$, we extract valid points using Eq. 1, yielding the final point cloud $\hat{\mathbf{V}}_{\text{pred}} \in \mathbb{R}^{N \times 3}$ and $\hat{\mathbf{V}}_{\text{gt}} \in \mathbb{R}^{N \times 3}$. The Chamfer distance is computed as

$$
\begin{aligned}
d(\hat{\mathbf{V}}_{\text{pred}}, \hat{\mathbf{V}}_{\text{gt}}) = & \frac{1}{2|\hat{\mathbf{V}}_{\text{pred}}|} \sum_{x \in \hat{\mathbf{V}}_{\text{pred}}} \min_{y \in \hat{\mathbf{V}}_{\text{gt}}} \|x - y\|_2 \\
& + \frac{1}{2|\hat{\mathbf{V}}_{\text{gt}}|} \sum_{y \in \hat{\mathbf{V}}_{\text{gt}}} \min_{x \in \hat{\mathbf{V}}_{\text{pred}}} \|x - y\|_2.
\end{aligned}
\tag{9}
$$

The F-score computes a harmonic mean of precision and recall regarding the two point clouds

$$
\text{Precision} = \frac{|\{x \in \hat{\mathbf{V}}_{\text{pred}} \mid \exists y \in \hat{\mathbf{V}}_{\text{gt}}, \|x - y\|_2 < \tau\}|}{|\hat{\mathbf{V}}_{\text{pred}}|},
\tag{10}
$$

$$
\text{Recall} = \frac{|\{y \in \hat{\mathbf{V}}_{\text{gt}} \mid \exists x \in \hat{\mathbf{V}}_{\text{pred}}, \|x - y\|_2 < \tau\}|}{|\hat{\mathbf{V}}_{\text{gt}}|},
\tag{11}
$$

$$
\text{FS}@\tau = \frac{2 \cdot \text{Precision} \cdot \text{Recall}}{\text{Precision} + \text{Recall}},
\tag{12}
$$

where $\tau$ is the threshold and is set to $0.1, 0.05, 0.02$ in this paper. As we compare between mesh-based and point-based methods, we perform sub-sampling for each method and the ground truth to ensure the number of points is the same.

**Object-level evaluation protocol**. To evaluate methods with an unspecified coordinate system or to evaluate under an unknown canonical coordinate system, we perform the brute-force search combined with the Iterative Closest Point (ICP) for point cloud registration. Specifically, we first translate the prediction by its averaged distance to the ground truth, then search for the minimal CD loss by transforming the prediction using 1000 candidate rotation angles (even partitioned for each axis). We further perform ICP to the rotation with minimal CD to optimize the pose. Despite becoming a convention

for object-level evaluation protocol, we argue that this approach is more similar to a workaround for current view-unaligned models. It is highly non-trivial to precisely align the point clouds by increasing the brute-force search iterations, and the process is inefficient as well.

### A.3. Seperate Encoders v.s. Shared Encoders

We compare our model's performance when using separate encoders versus a shared encoder for both point map prediction and stopping index prediction. Results on the SCRREAM dataset (Table 8) show that using a shared encoder performs reasonably but is consistently inferior to the separate encoder design. This is because LaRI's two prediction tasks are less coupled, leading to a competition between tasks. Therefore, we adopt separate encoders for improved accuracy. Nevertheless, in scenarios with strict computational or memory constraints, a shared encoder may remain a practical alternative with fewer parameters.

Table 8. Ablations of using separate encoders and one shared encoder.

| Method | CD↓ | FS@0.1↑ | FS@0.05↑ |
|---|---|---|---|
| Separate Encoder | **0.025** | **0.966** | **0.894** |
| Shared Encoder | 0.029 | 0.960 | 0.847 |

### A.4. Evaluation of Per-layer Predictions

To evaluate the performance gap between the visible layer and the occluded layers, as well as to investigate the model's performance under severe occlusions, we evaluate the per-layer reconstruction accuracy of LaRI. As shown in Table 9, the performance declines reasonably as the layer ID increases, since prediction through heavy occlusion becomes more challenging due to decreased context. The last layer is an exception because its valid intersecting area is very limited.

Table 9. **Per-layer prediction accuracy.** Quantitative results for different predicted layers on object-level (GSO) and scene-level (SCREAM) datasets.

| Layer ID | GSO (object-level) | | SCREAM (scene-level) | |
|---|---|---|---|---|
| | CD ↓ | FS@0.05 ↑ | CD ↓ | FS@0.05 ↑ |
| 1 | **0.023** | **0.890** | **0.057** | **0.588** |
| 2 | 0.033 | 0.809 | 0.089 | 0.426 |
| 3 | 0.031 | 0.796 | 0.102 | 0.358 |
| 4 | 0.032 | 0.792 | 0.104 | 0.374 |
| 5 | **0.023** | 0.835 | 0.097 | 0.408 |

### A.5. Comparison between point maps and depth maps

We compare LaRI's point map output with the depth map output used in generic LDIs (Shade et al., 1998). Although the two choices are theoretically equivalent, we find that estimating point maps yields better zero-shot generalization in 3D quality (as shown in Table 10) than estimating depth and focal length with separate heads. The performance gap stems from the ambiguity of focal-length estimation, which also appears in traditional depth estimation approaches (Facil et al., 2019; Yin et al., 2021) trained on larger datasets.

Table 10. **Comparison of different output representations.**

| Output type | CD ↓ | FS@0.05 ↑ |
|---|---|---|
| Depth map and focal length | 0.069 | 0.529 |
| Point map | **0.059** | **0.590** |

### A.6. More Visualizations

We show more results of the proposed model, including object-level results as in Fig. 7 and scene-level results as in Fig. 8, all shown in a multi-view manner to better identify unseen reconstruction performances. LaRI is capable of reconstructing single object occlusions (as in Fig. 7), scene foreground object occlusions, including the bed and the cabinet, as well as unseen room space reasoning, as shown in the last image of Fig. 8

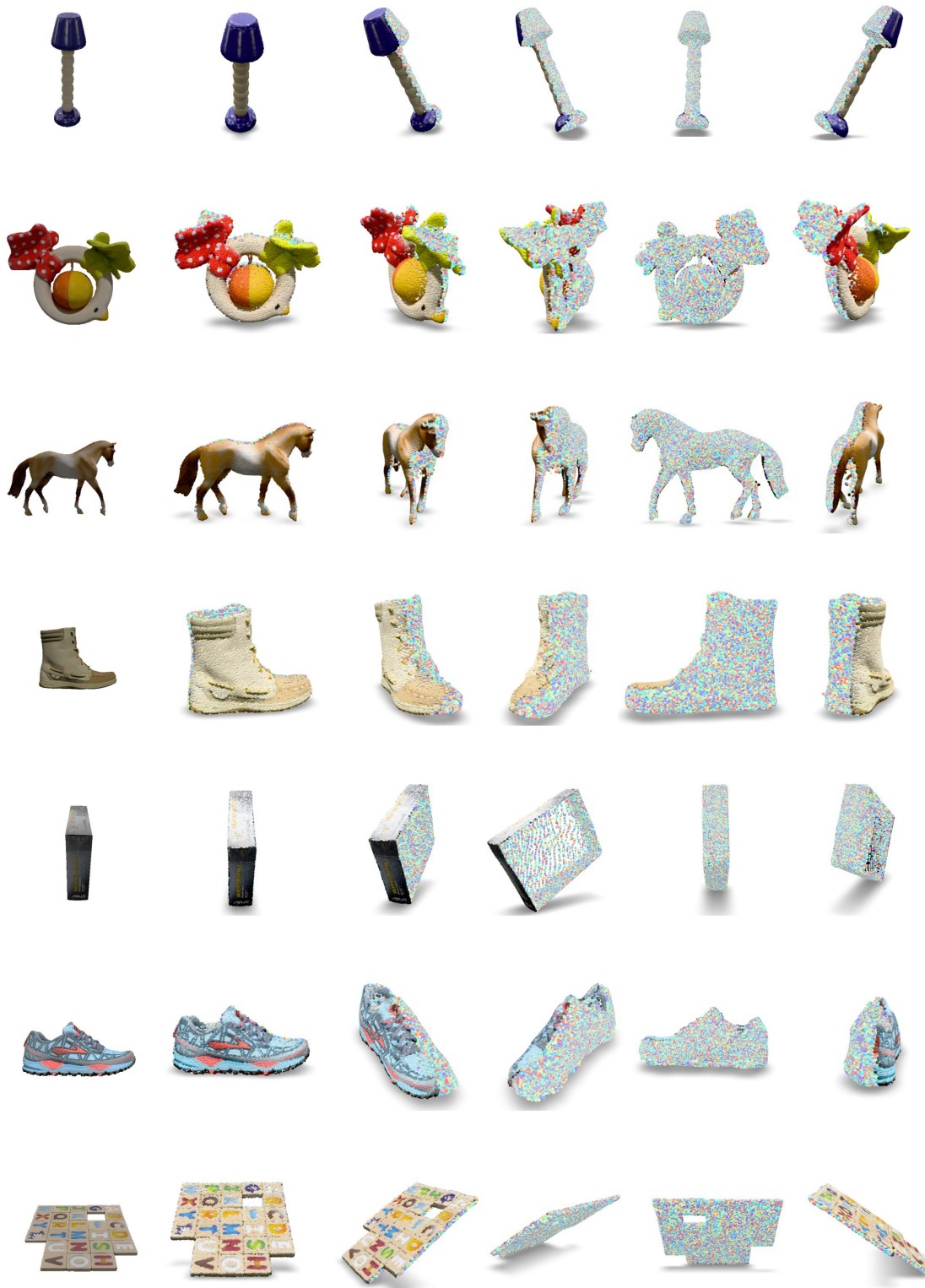

*Figure 7.* **Qualitative results of LaRI on GSO**. The leftmost image is the input, and the following images are LaRI's predicted models in multiple views. Random colors indicate the unseen parts.

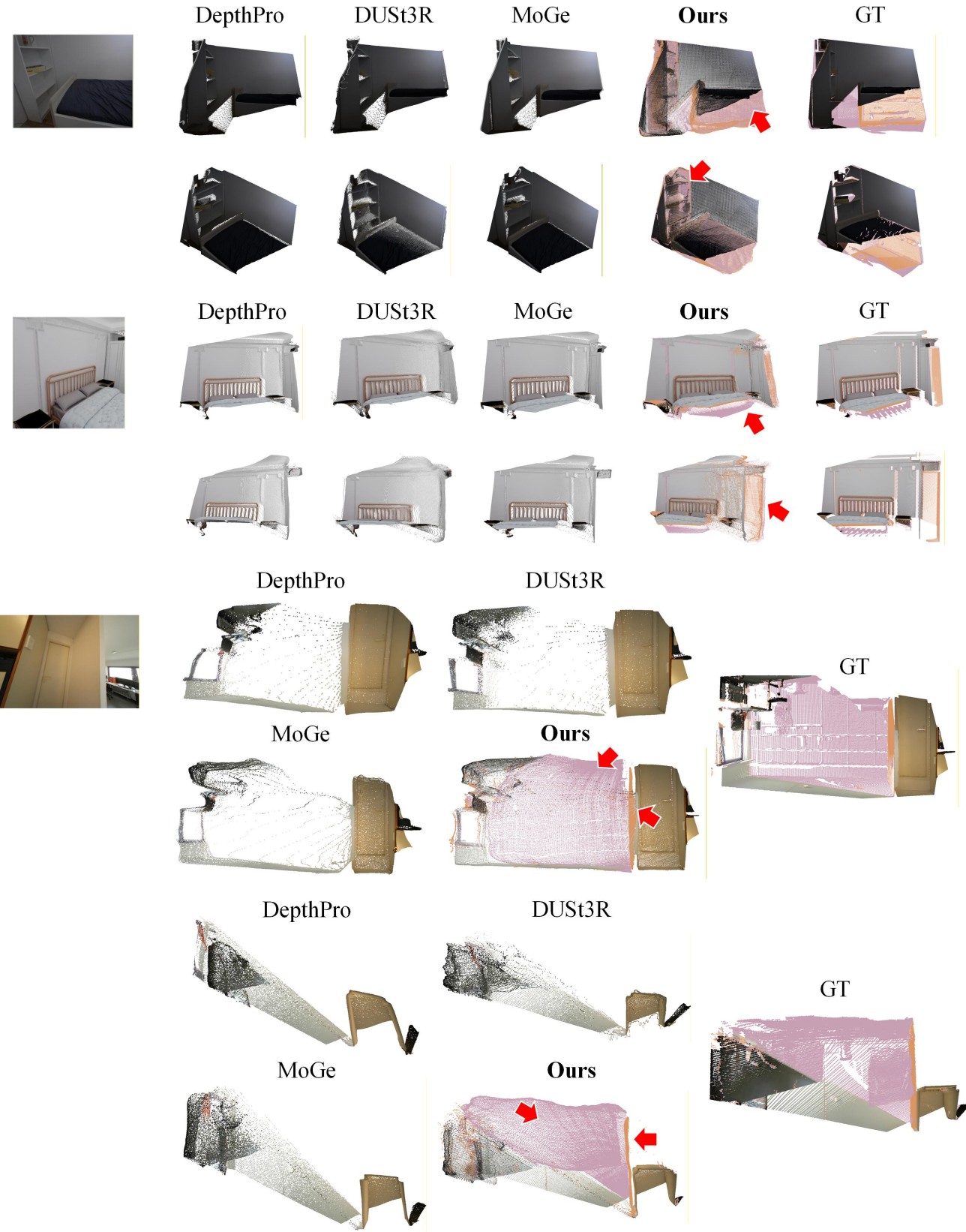

*Figure 8.* **Qualitative results of LaRI on indoor scenes**. From top to bottom: SCRREAM, 3D-FRONT, ScanNet++.

