# OpenReview forum: "LaRI: Layered Ray Intersections for Single-view 3D Geometric Reasoning"
_ICML.cc/2026/Conference — ICML 2026 regular_

### Official Review · Reviewer_6bxm · 2026-03-05

**Soundness:** 3
**Presentation:** 3
**Significance:** 2
**Originality:** 2
**Overall Recommendation:** 4
**Confidence:** 4

**Summary:**

In this paper, the authors propose Layered Ray Intersections (LaRI) to reason the occluded geometry from a single image. The model is supervised learned from several annotated datasets. With only one forward pass, LaRI can predict the occluded geometry for an object or a scene. A proposed ray-stopping index network make the predictions more meaningful. The visualization results show the ability of the model to reason the occluded part.

**Compliance With Llm Reviewing Policy:**

Affirmed.

**Final Justification:**

I would like to thank the authors for the detailed and comprehensive rebuttal. I appreciate the effort put into addressing my concerns, particularly in running the new evaluations. I would like to raise my rating to weak accept.

**Key Questions For Authors:**

Please refer to the weakness part.

**Limitations:**

Yes.

**Strengths And Weaknesses:**

Strengths:
1. The paper is well organized and easy to follow.
2. The annotation pipeline of the training data is valuable. This is helpful in the research area that reasoning the occluded geometry.
3. The model is unified for object-level and scene-level.

Weaknesses and questions:
1. Although the model is simple; however, most of the pipeline has been proposed, such as using the pointmap representation and using scale and shift to compute the loss, which limits the novelty of the paper.
2. The model limits the prediction of the number of occluded layers, which may mean it cannot handle an object or a scene with complex internal structure.
3. I would like to know why the authors using the pointmap as a representation of occluded geometry. What is the difference between using the pointmap or using a depth image as the representation?
4. I would like to know the performance of the model if the object or the scene has a structure that the occluded surface is close to the visible surface.

---

> ### Author Rebuttal · Authors · 2026-03-31
>
> We sincerely thank the reviewer for the constructive comments. Please find below our detailed responses and the modifications we will make in the final paper.
>
> > **Highlight the novelty.**
>
> Thank you for the comment. Rather than introducing new modules, LaRI’s novelty lies in a new _formulation and designs_, bringing _new capabilities and technical choices_ for unseen geometry estimation. Meanwhile, new _datasets and benchmarks_ are proposed to demonstrate this capability.
>
> 1. _Novel problem formulation._ LaRI formulates unseen geometry reasoning as a reverse process of ray-based rendering, defining a different formulation from generative, occupancy-based, and visible-surface-only approaches, bringing on-par or higher accuracy with better efficiency.
>
> 2. _New design._ LaRI proposes a novel ray-stopping index regression scheme to extract valid ray-surface intersections in this reverse formulation, which is more effective than mask regression by meaningful margins.
>
> 3. _New design choice._ LaRI shows that unseen geometric reasoning can be achieved not only through multiple inference steps or complex network structures, but also through minor adaptations to a feed-forward reconstruction network. This provides a new design choice for future works.
>
> 4. _New datasets._ As a non-technical contribution, LaRI proposes a full data annotation pipeline, together with available data, evaluation benchmarks, and tools to support further data expansion in future work.
>
> > **Limited number of occluded layers may limit the handling of complex structures.**
>
> LaRI’s _redundant yet compact_ layered output together with _stopping index regression_ enables flexible handling of complex structures.
>
> First, using more redundant layers allows LaRI to represent more complex structures, while the stopping index regression flexibly extracts both simple and complex structures from redundant layers.
>
> Second, due to the compactness of LaRI’s output, setting the number of layers above 5 already provides nearly complete structural coverage on _all_ datasets, as computed from the GT below. This supports the sufficiency of the current setting on _existing_ datasets, while the handling of even more complex structures will be further explored in future work.
>
> | **Layer Number (Scene)** | **1** | **3** | **5** | **7** | **9** |
> |---|---:|---:|---:|--:|--:|
> | 3D-FRONT | 42.2% | 73.9% | 89.6% | 96.3% | 99.3% |
> | SCRREAM | 50.7% | 71.9% | 98.3% | 99.8% | 99.9% |
> | ScanNet++ | 74.0% | 94.1% | 98.4% | 99.6% | 99.9% |
>
> | **Layer Number (Object)** | **1** | **3** | **5** | **7** | **9** |
> |--|--:|--:|--:|--:|---:|
> | GSO | 41.0% | 89.5% | 99.4% | 99.6% | 99.9% |
> | Objaverse | 48.3% | 94.5% | 99.7% | 99.8% | 99.9% |
>
> > **Why not use depth for layered output, and what is the difference between using point maps and depth?**
>
> The main difference is that point maps implicitly encode focal length in the XYZ coordinates, which we find _generalizes better_ than separately predicting depth and focal length.
>
> To obtain a 3D point cloud as the final output, we can either predict point maps in XYZ coordinates or predict depth (the Z coordinate) together with focal length, and then derive the corresponding X and Y coordinates. Although the two are theoretically equivalent, we empirically find that estimating point maps yields better zero-shot generalization in 3D quality (as shown below) than estimating depth and focal length with separate heads. The performance gap stems from the ambiguity of focal-length estimation, which also appears in traditional depth estimation approaches [1,2] trained on larger datasets. We will add more clarifications in the revised paper.
>
> | **Results on SCRREAM** | **CD↓** | **FS@0.5↑** |
> |---|---:|---:|
> | Depth map and focal length | 0.069 | 0.529 |
> | Point map | **0.059** | **0.590** |
>
> > **Model performance on occluded structures that are close to the visible surface.**
>
> Following the reviewer’s suggestion, we evaluate the model by progressively incorporating occluded structures in both GT and predictions that lie within a distance threshold behind the visible surface. The threshold is defined as a ratio (%) with respect to the GT distance discrepancy of the scene. We choose small distance ratios (5%–20%) to extract occluded surfaces _near_ the visible surface.
>
> The results show that the model performs reasonably well in near-surface areas, benefiting from the richer context provided by visible regions. We will update the results and add more discussions in the final paper.
>
> |  | **GSO (object-level)** |  | **SCRREAM (scene-level)** |  |
> |---|---:|---:|---:|---:|
> | **Distance Ratio** | **CD↓** | **FS@0.05↑** | **CD↓** | **FS@0.05↑** |
> | 5% | 0.022 | 0.896 | 0.054 | 0.612 |
> | 10% | 0.021 | 0.903 | 0.053 | 0.622 |
> | 20% | 0.021 | 0.909 | 0.054 | 0.614 |
>
> [1] CAM-Convs: Camera-Aware Multi-Scale Convolutions for Single-View Depth. CVPR 2019.
>
> [2] Learning to Recover 3D Scene Shape from a Single Image. CVPR 2021.

---

> > ### Author Rebuttal · Reviewer_6bxm · 2026-04-03
> >
> > I would like to thank the authors for the detailed and comprehensive rebuttal. I appreciate the effort put into addressing my concerns, particularly in running the new evaluations. I would like to raise my rating to weak accept.

---

> > > ### Author Response · Authors · 2026-04-03
> > >
> > > Thank you very much for your thoughtful follow-up and for taking the time to carefully evaluate our rebuttal. We sincerely appreciate your recognition of our additional experiments and clarifications.
> > >
> > > We will incorporate the new evaluations and clarifications discussed here to further strengthen the paper.

---

### Official Review · Reviewer_S5UE · 2026-03-11

**Soundness:** 3
**Presentation:** 3
**Significance:** 3
**Originality:** 3
**Overall Recommendation:** 4
**Confidence:** 4

**Summary:**

This work propose a layered point map representation to describe 3D objects or scenes, which is able to complete missing parts compared with single depth representation. Despite it shows effectiveness, training data is still one of the biggest concern.

**Compliance With Llm Reviewing Policy:**

Affirmed.

**Final Justification:**

While I still think the connection to LDI should be discussed more explicitly, I acknowledge that the authors' contribution goes beyond simply applying LDI to reconstruction. The authors' rebuttal adequately addressed my data accessibility concern with concrete references to frame accumulation strategies, and they agreed to discuss the synthetic-real domain gap more thoroughly.

Given that this is among the first works to bring multi-layer geometric prediction into a feed-forward reconstruction setting, and the experimental results are solid, I'm raising my score to 4 (weak accept).

**Key Questions For Authors:**

Please justify your method with the weakness stated above.

**Limitations:**

yes

**Strengths And Weaknesses:**

Strengths:
1. the representation is simple and can infer occluded geometry from single image.
2. Experiments show competitive results and good efficiency.

Weakness:
1. this representation seems very similar to layered depth representation, which has been explored in the literature. so the novelty is unclear. Layered depth image is a old and well-known representation in graphics which can date back to 1998: https://dl.acm.org/doi/10.1145/280814.280882. So the representation itself of this paper shouldn't be considered as one novelty.

2. training data is actually hard to access. For real world cases, it's pretty hard to get all intersection points of a ray, making the representation limit to toy setting. Data being hard to access is a big limitation. The first-layer depth information could be accessed through some lidar sensor, but it's much harder to get the following intersection points in real world cases. This limitation is also pointed and discussed in the limitation section of this paper.

---

> ### Author Rebuttal · Authors · 2026-03-31
>
> We thank the reviewer for the constructive comments. Please find below our detailed responses and the changes we will make in the final paper.
>
> > **Highlight the novelty, particularly regarding the LDIs in graphics.**
>
> While we agree that recording layered depth images (LDIs) in _graphics rendering_ with _known 3D assets_ has been explored, LaRI’s novelty lies in transforming this rendering concept into a complete _reverse process_, i.e., a fully supervised image-to-3D reconstruction task, with _new formulations_, _new datasets, data generation pipelines, and benchmarks_, rather than a naive reuse of existing LDI concepts:
>
> 1. _New formulation._ LaRI fundamentally differs from prior graphics works in three aspects. (1) _Target_: LDIs in graphics aims at efficiency in storage and _image_ quality in rendering. By contrast, LaRI targets _estimating 3D geometry_ in high quality. Therefore, the design choices are different, as for estimation, the most compact data structure is often not the best. We do not emphasize point merging and color interpolation like traditional LDIs or Surfels. Similar to the seminal paper DUST3R, we estimate redundant point maps, rather than just depth information. We also propose new data-ordering strategies that are beneficial for estimation but not necessary for rendering. (2) _Pipeline_: graphics approaches use layered depth only as view-warping _intermediates_ in a physical rendering pipeline, whereas LaRI treats layered point maps as the _final neural prediction targets_. (3) _Input assumptions_: graphical layered depth assumes known 3D geometry as input, whereas LaRI takes an image as input and predicts unknown geometry.
>
> 2. _New designs._ Unlike graphics methods, where ray-surface intersections can be directly computed from known 3D geometry, reconstruction approaches must estimate valid intersections to recover the final shape. LaRI therefore proposes a novel design to identify valid ray-surface intersections via ray-stopping index regression, which is validated to be more effective than the commonly used mask-based regression.
>
> 3. _Novel data generation pipeline and datasets._ We contribute a new data generation pipeline and new datasets for this underexplored task. The pipeline and datasets cover various data formats, bridging the gap needed to train the above reverse-formulation model.
>
> We will add more detailed discussions of the similarities and differences between the two schemes to improve clarity.
>
> > **Training data is hard to acquire in real-world cases.**
>
> Training data access _is not theoretically restricted, but it does require additional engineering effort_. Specifically, the training data can be scaled through both extensive real-world first-layer annotations and synthetic data.
>
> 1. _Frame accumulation for real-world data generation._ Multiple first-layer scans (e.g., per-view GT depth) in a scene can be accumulated into complete scene-level geometry for multi-layer annotation, avoiding the need to acquire all intersections in a single scan. One can rely on:
>    1. Abundant existing datasets, with tolerance to some noise and incompleteness, which provide rich and diverse real-world scenarios up to millions of images. This approach has been widely used in scene reconstruction [1] and occupancy prediction tasks [2,3,4], covering both indoor and outdoor datasets.
>    2. Task-dedicated scans to cover regions that previous datasets may overlook, providing higher data quality and scene coverage at moderate additional cost by more diverse viewpoint sampling. We aim to build a series of dedicated datasets in future work.
>
> 2. _Incorporating synthetic data._ Previous research [5] highlights the effectiveness of _synthetic_ data for _real-world_ 3D perception. Therefore, large-scale 3D object datasets provide an alternative path for rapid scaling with _clean_ geometry: (1) scenes can be composed with multiple 3D objects, including background objects, using graphics engines, and then rendered together with multi-layer annotations; (2) existing pipelines such as InfiGen [6] can also generate diverse synthetic scenes. Considering the number of available objects (up to several millions) and their possible combinations, the dataset scale can be meaningfully expanded.
>
> Overall, there remains ample room for dataset expansion, and we will continue _expanding_ existing datasets and _creating dedicated_ datasets to support future research.
>
> [1] Continuous 3D Perception Model with Persistent State. ICCV 2025.
>
> [2] Behind the Scenes: Density Fields for Single View Reconstruction. CVPR 2023.
>
> [3] SurroundOcc: Multi-Camera 3D Occupancy Prediction for Autonomous Driving. ICCV 2023.
>
> [4] Know Your Neighbors: Improving Single-View Reconstruction via Spatial Vision-Language Reasoning. CVPR 2024.
>
> [5] Repurposing Diffusion-Based Image Generators for Monocular Depth Estimation. CVPR 2024.
>
> [6] Infinite Photorealistic Worlds using Procedural Generation. CVPR 2023.

---

> > ### Author Rebuttal · Reviewer_S5UE · 2026-04-04
> >
> > Thanks for the reply. My second concern is mainly solved but I have some following questions about the first point.
> >
> > For novelty over LDI, I don't agree with the author that your paper propose a new formulation. The paper is more of applying existing LDI representation in reconstruction task. The "New formulation" and "New data generation pipeline and datasets" are answering: How you apply existing LDI representation in reconstruction task, which doesn't relate with the representation itself. The paper should position its contribution accordingly, which seems requiring substantial reframing. Do you agree with above?
> >
> > For training data, I do agree multiple first-layer scans can be accumulated into scene-level geometry in some certain cases. In rebuttal the authors mentioned four works in this line, which seems reasonable and address my concern accordingly. For the synthetic data, I think there's still domain gap between synthetic and real scans, which doesn't seems helpful for reconstructing real world with multiple layers. I suggests that authors should discuss this data related concern more seriously in the paper.

---

> > > ### Author Response · Authors · 2026-04-07
> > >
> > > We thank the reviewer for the insightful and constructive follow-up.
> > >
> > > > **Regarding the LDIs.**
> > >
> > > We _agree_ that LaRI is related to the layered-depth image data structure in graphics (e.g., LDIs and surfels). However, we clarify that LaRI’s contribution is not a plain adoption of LDIs. We will _revise the paper_ to make this connection clearer, but we do not see this as requiring substantial reframing.
> > >
> > > **Clarification of contributions.** We didn't claim that LaRI's layered storage primitive itself is unique. Rather, the major contribution of LaRI is to frame _single-view occluded 3D geometry reasoning_ as a _supervised layered ray-intersection prediction problem_, and to present a practical solution that makes it _learnable and effective_. This requires multiple distinct ingredients beyond the seminal LDIs, including layered point maps, ray-stopping index regression, data generation, and benchmarking. The contribution is thus systematic and more comprehensive than an isolated comparison of a seminal paper serving a substantially _different_ purpose.
> > >
> > > **Comparing specifically with LDIs.** When setting aside the differences in task and pipeline, we _agree_ on the similarity to the layered-depth image data structure (e.g., LDIs, surfels) in the graphics domain. We will add more discussion to clarify the connection, as follows. Apart from that, LaRI's output is not identical to classical LDI, as it predicts 3D point maps instead of depth maps, thereby changing the inductive bias from image-based rendering to camera-aligned 3D reconstruction and leading to _better generalization_ of 3D quality, as shown below.
> > >
> > > | **Results on SCRREAM** | **CD↓** | **FS@0.05↑** |
> > > |---|---:|---:|
> > > | Depth map | 0.069 | 0.529 |
> > > | Point map | **0.059** | **0.590** |
> > >
> > > **Revision of the paper.** We aim to address the reviewer's concern with the following revisions:
> > >
> > > 1. Clarify the historical connection to LDIs in the Introduction (Section 1) and Method (Section 3.1);
> > > 2. Expand the discussion of LDIs and related layered graphics representations in Related Work (Section 2);
> > > 3. Add experimental results (Section 4) further justifying the methodological designs in LaRI, including point-map prediction and redundant layers with stopping-index regression.
> > >
> > > We _agree_ to clarify the paper's positioning with respect to LDIs more explicitly. However, we _do not see this as requiring substantial reframing_, because the core claimed contributions already lie on the methodological level, focusing on _a single-view occluded 3D geometry reasoning model_, rather than a claim of an entirely new concept. In addition to the above revisions, we will refine the wording throughout the paper to better address the reviewer's comments and to avoid possible misunderstandings about the paper’s claimed contributions.
> > >
> > > > **Regarding Datasets.**
> > >
> > > We are encouraged that the accumulation of multi-layer scans has addressed the reviewer's concerns about real-world data. We will continue expanding the datasets using both existing and newly collected datasets to foster future research.
> > >
> > > For synthetic data, we also _agree_ that there is a gap between synthetic and real-world data. We will address the reviewer's concern by discussing this limitation in the data-related sections (Section 4.1, Appendix A.1) and the Limitations Section (Section 5).

---

### Official Review · Reviewer_fYBa · 2026-03-13

**Soundness:** 4
**Presentation:** 4
**Significance:** 3
**Originality:** 4
**Overall Recommendation:** 5
**Confidence:** 4

**Summary:**

I really like this work. LaRI introduces a layered representation for 3D geometry that effectively addresses longstanding limitations in prior representations, such as the inability to reconstruct occluded (unseen) regions and the excessive redundancy often seen in methods like NeRF. By endowing reconstruction pipelines with generative-model-like capabilities—particularly the ability to hallucinate plausible geometry in occluded areas—LaRI represents a meaningful step forward in single-view 3D geometric reasoning.

**Compliance With Llm Reviewing Policy:**

Affirmed.

**Final Justification:**

Thanks to the author, my concerns have been addressed

**Key Questions For Authors:**

- Will the authors open-source the dataset, trained models, code, and (ideally) the full data generation pipeline? Releasing these resources would have substantial impact and enable easier reproducibility and extension by the community.
- What is the actual prediction accuracy of the stop index map, particularly in cases with more than 3 intersection layers? An analysis breaking down per-layer prediction precision (e.g., accuracy for the first, second, third, and higher-order layers) would provide deeper insight into the representation's strengths and limitations in handling complex occlusion structures.

**Limitations:**

yes

**Strengths And Weaknesses:**

#### Strengths

- The paper proposes a novel modeling approach based on Layered Ray Intersection Maps, which differs fundamentally from conventional light-transport or graphics-pipeline paradigms. By predicting a stop index map to determine the number of ray-surface intersections per pixel, the representation achieves compactness while enabling meaningful completion/filling of occluded regions. This design is highly innovative and elegant.
- The authors adopt a feed-forward paradigm aligned with current trends in the field, resulting in strong generalization and high inference efficiency.
- The creation of a large-scale dataset tailored to this representation is a significant contribution that paves the way for future research in layered or point-map-based 3D reasoning.
- The demonstrated ability to predict plausible geometry in unseen/occluded regions is highly convincing and serves as strong evidence of the method's capabilities.
- Both quantitative and qualitative results at the object level show performance comparable to—or in some cases surpassing—that of dedicated generative models.
- The ablation studies are comprehensive and clearly demonstrate the contribution of each module and design choice.

#### Weaknesses

- Reconstruction accuracy in visible regions remains somewhat limited. This may be attributable to dataset scale or diversity, but it would strengthen the evaluation to include stronger recent feed-forward baselines such as VGGT (Visual Geometry Grounded Transformer) for direct comparison.
- The current model is restricted to single-image input and lacks multi-view aggregation or camera pose estimation capabilities, which limits its applicability in more realistic capture scenarios.
- typo error: line 157-158 AModelDepth. line 260-261 reference error.

---

> ### Author Rebuttal · Authors · 2026-03-31
>
> We truly appreciate the reviewer’s constructive and insightful comments, which help further improve the paper. Please find below our responses and the modifications we will make in the final paper.
>
> > **Strengthen the evaluation by adding recent feed-forward approaches such as VGGT.**
>
> Thank you very much for pointing this out. The performance gap on visible surfaces mainly stems from the evaluation protocol, training data scale, and supervision signals. (Due to the character limit, please refer to Answer 2 to R1.) We will incorporate the updated results and add more illustrations for clarity.
>
> We appreciate the reviewer’s comment and have added the results of VGGT on both object-level and scene-level data. VGGT falls behind generative approaches (e.g., TRELLIS) and our method on object-level data, while performing on par with or slightly worse than the single-view method MoGe, on scene-level data. We will update the results in the revised paper.
>
> | **Object Results** | **CD↓** | **FS@0.1↑** | **FS@0.05↑** | **FS@0.02↑** |
> |---|---:|---:|---:|---:|
> | VGGT | 0.332 | 0.335 | 0.274 | 0.200 |
> | TRELLIS | 0.027 | 0.959 | 0.853 | 0.608 |
> | MoGe | 0.160 | 0.373 | 0.275 | 0.192 |
> | Ours | 0.025 | 0.966 | 0.894 | 0.643 |
>
> | **Scene Results** | **Visible CD↓** | **Visible FS@0.05↑** | **Unseen CD↓** | **Unseen FS@0.05↑** | **Overall CD↓** | **Overall FS@0.05↑** |
> |---|---:|---:|---:|---:|---:|---:|
> | MoGe | 0.035 | 0.786 | - | - | 0.063 | 0.668 |
> | VGGT | 0.040 | 0.745 | - | - | 0.068 | 0.637 |
> | Ours | 0.057 | 0.589 | 0.077 | 0.494 | 0.059 | 0.590 |
>
> > **The single-view model is limited in processing multi-view inputs.**
>
> Thank you for the insightful comment. Geometric perception models (e.g., depth estimation) generally evolve from single-view to multi-view settings, so that valid design choices from predecessor models can be reused or extended. As LaRI is still at an early stage of this line of exploration, we would be very glad to extend LaRI to multi-view models in future work.
>
> > **Open-source of the dataset, model weights, code, and ideally the generation pipeline.**
>
> Thank you for the highly constructive comment. We will open-source the full data generation pipeline with clear instructions, the full datasets used to train our model, the full pre-trained weights, and the training/evaluation code once the paper is finalized. We are delighted to see further technical developments and progressive dataset extensions that can foster broader capabilities in geometric perception models.
>
> > **Breaking down per-layer predictions for a more informed evaluation.**
>
> Thank you for the constructive comment. We add per-layer evaluation in the following table for both object-level and scene-level data. The performance declines reasonably as the layer ID increases, since prediction through heavy occlusion becomes more challenging due to decreased context. The last layer may lead to biased estimation because its area of valid intersecting regions is limited for both predictions and GT. We will update the results and analysis in the final paper.
>
> |  | **GSO (object-level)** |  | **SCRREAM (scene-level)** |  |
> |---|---:|---:|---:|---:|
> | **Layer ID** | **CD↓** | **FS@0.05↑** | **CD↓** | **FS@0.05↑** |
> | 1 | 0.023 | 0.890 | 0.057 | 0.588 |
> | 2 | 0.033 | 0.809 | 0.089 | 0.426 |
> | 3 | 0.031 | 0.796 | 0.102 | 0.358 |
> | 4 | 0.032 | 0.792 | 0.104 | 0.374 |
> | 5 | 0.023 | 0.835 | 0.097 | 0.408 |
>
> > **Revise some typos.**
>
> Thank you for pointing this out. On line 157, we will revise the method name to “Amodal Depth Anything,” following the official naming in the original paper [1]. Meanwhile, on lines 260–261, we will revise the citation of model weights to the DINO series [2,3] and MoGe [4], which are used in different experiments.
>
> [1] Amodal Depth Anything: Amodal Depth Estimation in the Wild. ICCV 2025.
>
> [2] DINOv2: Learning Robust Visual Features without Supervision. TMLR 2024.
>
> [3] DINOv3. arXiv 2025.
>
> [4] MoGe: Unlocking Accurate Monocular Geometry Estimation for Open-Domain Images with Optimal Training Supervision. ICCV 2025.

---

### Official Review · Reviewer_6bA6 · 2026-03-19

**Soundness:** 3
**Presentation:** 2
**Significance:** 3
**Originality:** 3
**Overall Recommendation:** 5
**Confidence:** 4

**Summary:**

This paper aims to recover not only the front but also occluded scene and object structure along each camera ray. To address this, the authors propose LaRI that predicts multiple ordered 3D intersection points per pixel, together with a ray stopping index model to determine the valid number of intersections along each ray. Experiments on both object-level and scene-level benchmarks show that the method is especially effective for unseen surface reasoning, achieves strong overall reconstruction quality, and offers a favorable efficiency-accuracy tradeoff.

**Compliance With Llm Reviewing Policy:**

Affirmed.

**Final Justification:**

I appreciate the authors' detailed clarifications and additional experiments in the rebuttal. All of my concerns have been well addressed. I am willing to raise my score from weak accept to accept.

**Key Questions For Authors:**

I am curious about the result in Table 8. Intuitively, it would be better to include a shared encoder for the joint optimization of point map decoder and index decoder. Why does the result of separate encoder show even better performance?

**Limitations:**

yes

**Strengths And Weaknesses:**

**Strengths**

1.	The core idea of learning multi-layer point maps is interesting and novel, and the experimental design is thoughtful and well motivated.

2.	The ablation studies are comprehensive. The paper ablates each major component of the method as well as relevant hyperparameters, and the results demonstrate their effectiveness.

3.	The paper is well-written and clear to understand.

**Weaknesses**

1.	The experimental settings are somewhat unusual. Table 2 reports comparisons with 3D generation methods on object-level datasets, whereas Table 3 reports comparisons with depth prediction methods on scene-level datasets. Although I understand that 3D scene generation methods are still relatively rare, Table 2 should at least also include comparisons with these depth prediction methods.

2.	In Table 3, LaRI performs worse than MoGe in the visible region, where the latter also adopts a ViT encoder+CNN decoder architecture. Does this suggest that predicting multi-layer point maps may compromise the model’s spatial perception ability? More discussion is needed here.

3.	The method successfully predicts multiple intersections behind the visible surfaces. However, based on the visual results, the reconstruction quality does not appear very strong. In indoor scenes especially, the predicted content seems to deviate noticeably from the ground truth.

---

> ### Author Rebuttal · Authors · 2026-03-31
>
> We sincerely thank the reviewer for the thoughtful and highly constructive comments. Please find below the responses and the changes we will make to the final paper.
> > **Compare with depth prediction methods in object-level reconstruction (Table 2).**
>
> We thank the reviewer for the suggestion. We re-evaluate depth prediction methods, together with VGGT, on object-level reconstruction under the view-aligned setting, since their outputs are inherently view-aligned. These methods perform notably worse than object-level methods (e.g., TRELLIS) and our approach due to their incomplete geometry. We will add this comparison to Table 2 in the final version.
>
> | **Method** | **CD↓** | **FS@0.1↑** | **FS@0.05↑** | **FS@0.02↑** |
> |---|---|---|---|---|
> | TRELLIS | 0.027 | 0.959 | 0.853 | 0.608 |
> | Metric3Dv2 | 0.270 | 0.330 | 0.204 | 0.092 |
> | DepthPro | 0.310 | 0.313 | 0.224 | 0.122 |
> | DUSt3R | 0.220 | 0.324 | 0.240 | 0.153 |
> | VGGT | 0.332 | 0.335 | 0.274 | 0.200 |
> | MoGe | 0.160 | 0.373 | 0.275 | 0.192 |
> | Ours | 0.025 | 0.966 | 0.894 | 0.643 |
>
>
> > **Discuss the performance gap with MoGe in visible regions despite similar network designs. Is it a performance compromise caused by multi-layer prediction?**
>
> The visible-region gap mainly comes from the _evaluation protocol_, _LaRI’s smaller training data_, and the _trade-off introduced by multi-layer training_.
>
> 1. _Evaluation protocol_. MoGe-like methods align predictions and GT only on the visible surface, whereas LaRI aligns across all layers to keep a unified protocol for visible/unseen/full evaluation. This propagates global alignment errors into the visible layer. Under MoGe’s alignment protocol, LaRI shows notably better results (rows 4 and 5). This empirically accounts for most of the gap, and we add more discussions in the final paper.
>
> 2. _Data scale_. As a preliminary model, LaRI is fine-tuned on around 200K images, much smaller in scale and diversity than MoGe’s over one million images, leading to weaker generalization. Fine-tuning MoGe on LaRI data gives results similar to LaRI trained on visible surfaces only (rows 2 and 3), and both are worse than the original MoGe (rows 1 and 2). This confirms the importance of data scale, and we will continue expanding LaRI’s dataset to foster future research.
>
> 3. _Effect of multi-layer training_. We re-train LaRI with visible-surface-only supervision and visible-surface-only alignment (row 3), following MoGe. Rows 3 and 4 show that multi-layer supervision slightly degrades performance, possibly because the loss is averaged across layers. We aim to address this issue in future work.
>
>
> |  | **SCRREAM Results (visible layer)** | **LaRI Data** | **Full-layer Align.** | **Multi-layer Super.** | **CD↓** | **FS@0.5↑** |
> |---|---|---|---|---|---|---|
> | 1 | MoGe |  |  |  | 0.035 | 0.786 |
> | 2 | MoGe (LaRI data fine-tuned) | Yes |  |  | 0.040 | 0.735 |
> | 3 | LaRI (visible layer supervision) | Yes |  |  | 0.042 | 0.732 |
> | 4 | LaRI (visible layer alignment) | Yes |  | Yes | 0.045 | 0.688 |
> | 5 | LaRI (Ours) | Yes | Yes | Yes | 0.057 | 0.589 |
>
> > **Visualization quality in unseen layers for indoor scenes.**
>
> Thank you for the insightful comment. Estimating unseen geometry under heavy occlusion is indeed challenging, as the available context decreases with layer index. Since the degradation remains relatively controlled, as shown below, and this method is still among the first few explorations in this direction, we will continue improving it in future work.
>
> |  | **GSO (object-level)** |  | **SCRREAM (scene-level)** |  |
> |--|--:|--:|--:|--:|
> | **Layer ID** | **CD↓** | **FS@0.05↑** | **CD↓** | **FS@0.05↑** |
> | 1 | 0.023 | 0.890 | 0.057 | 0.588 |
> | 2 | 0.033 | 0.809 | 0.089 | 0.426 |
> | 3 | 0.031 | 0.796 | 0.102 | 0.358 |
> | 4 | 0.032 | 0.792 | 0.104 | 0.374 |
> | 5 | 0.023 | 0.835 | 0.097 | 0.408 |
>
> > **Discuss why separate encoders for point map prediction and index prediction lead to better performance in Table 8?**
>
> Separate encoders provide _higher capacity_ and _reduce task competition_ in our setting.
>
> According to multi-task learning literature [1], a shared encoder helps when tasks are closely related and model capacity is sufficient. Unlike joint depth and surface normal estimation, where both outputs describe the same surfaces, or joint depth and semantic segmentation, where geometry and semantics are complementary, LaRI’s two subtasks are less coupled: the stopping index predicts the number of valid layers rather than category-level cues, and layered intersection coordinates are not directly informative for predicting the valid intersection count. As a result, the two tasks are not strongly aligned, so a shared encoder introduces more competition than benefit.
>
> We also _agree_ that a better shared-encoder design is an interesting direction to explore, and we leave it for future work.
>
> [1] Which Tasks Should Be Learned Together in Multi-task Learning? ICML 2020.

---

> > ### Author Rebuttal · Reviewer_6bA6 · 2026-04-04
> >
> > I appreciate the authors' detailed clarifications and additional experiments in the rebuttal. All of my concerns have been well addressed. I am willing to raise my score from weak accept to accept.

---

> > > ### Author Response · Authors · 2026-04-04
> > >
> > > Thank you very much for your thoughtful feedback and for taking the time to review our rebuttal. We sincerely appreciate your recognition of our clarifications. We will incorporate the new results and expanded discussion to further strengthen the paper.

---

### Decision · Program_Chairs · 2026-04-30

**Decision:**

Accept (regular)

**Comment:**

This submission received mixed reviews initially (4, 5, 3, 3). After the rebuttal and lengthy discussion, three reviewers increased their scores, resulting in final unanimously positive final scores of (5, 5, 4, 4). Concerns were raised in the reviews regarding limited performance gains, limited technical innovation over existing methods, and the limitations of the model design. During the discussion, the authors provided extensive new results and clarifications, which further reinforced the benefits of the proposed method. All reviewers find the arguments convincing and recommends accepting the submission. The Area Chairs agree with their assessment, and encourage the authors to incorporate the feedback into the revision.